# MIND YOUR BITS AND ERRORS: PRIORITIZING THE BITS THAT MATTER IN VARIATIONAL AUTOENCODERS

## ABSTRACT

Good likelihoods do not imply great sample quality. However, the precise manner in which models trained to achieve good likelihoods fail at sample quality remains poorly understood. In this work, we consider the task of image generative modeling with variational autoencoders and posit that the nature of high-dimensional image data distributions poses an intrinsic challenge. In particular, much of the entropy in these natural image distributions is attributable to visually imperceptible information. This signal dominates the training objective, giving models an easy way to achieve competitive likelihoods without successful modeling of the visually perceptible bits. Based on this hypothesis, we decompose the task of generative modeling explicitly into two steps: we first prioritize the modeling of visually perceptible information to achieve good sample quality, and then subsequently model the imperceptible information—the bulk of the likelihood signal—to achieve good likelihoods. Our work highlights the well-known adage that "not all bits are created equal" and demonstrates that this property can and should be exploited in the design of variational autoencoders.

## 1 INTRODUCTION

The task of generative modeling of high-dimensional image data has inspired the development of many successful deep generative models such as autoregressive models (Uria et al., 2016; Oord et al., 2016b), flow-based models (Dinh et al., 2014; 2016), generative adversarial networks (Goodfellow et al., 2014), variational autoencoders (Kingma & Welling, 2013; Rezende et al., 2014), and diffusion models (Ho et al., 2020; Song & Ermon, 2019). These models have since been applied to many other modeling tasks such as speech synthesis (Oord et al., 2016a; Donahue et al., 2018), reinforcement learning (Zhang et al., 2019; Levine et al., 2019), and scene-understanding (Eslami et al., 2018). In this work, we focus on the application of VAEs to image modeling. Despite the success of variational autoencoders (VAEs) in numerous fields (Akuzawa et al., 2018; Hsu et al., 2017; Gómez-Bombarelli et al., 2018; Sultan et al., 2018; Van Hoof et al., 2016), the application of VAEs to image modeling has been lukewarm at best—plagued by optimization issues (Rezende & Viola, 2018; Child, 2020; Vahdat & Kautz, 2020) and models achieving good Evidence Lower Bounds (ELBOs) but poor sample quality. Our work examines the phenomenon that good ELBOs do not impy great sample quality in VAEs (Theis et al., 2015). We argue that the primary cause of this issue stems from the model being overwhelmed by the vast volume of visually-imperceptible information contained in natural image distributions. Our contributions are thus as follows.

1. We analyze the rate (the amount of information encoded in the latent space) of VAEs trained with different rate-distortion trade-offs and show that low-rate models can achieve perceptually high-quality reconstructions and sampling. In contrast, the standard ELBO objective favors high-rate models with good reconstructions but poor sampling quality.

2. However, low-rate models have much worse ELBOs due to poor modeling of the visually imperceptible information. To overcome this issue, we propose a two-stage training process that trains a secondary high-rate model on top of the low-rate model. Since the secondary model is restricted to modeling visually imperceptible information, it can improve the ELBO significantly with minimal impact on the sample quality achieved by the initial low-rate model.

## 2 THE CHALLENGE OF TRAINING VARIATIONAL AUTOENCODERS

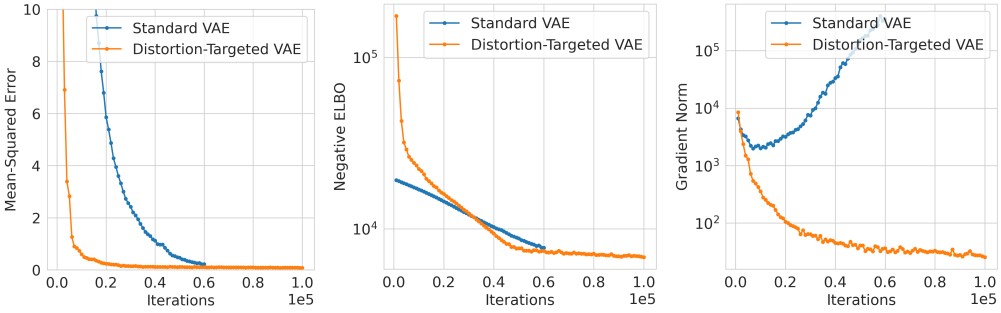

Figure 1: Mean-squared error, ELBO, and gradient norm for the standard VAE with trainable variance versus a VAE targeted to achieve the same distortion. The former exhibits unstable training.

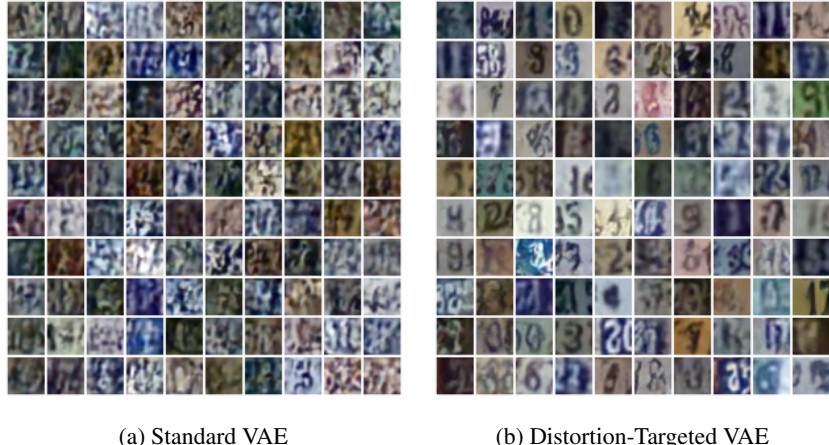

(a) Standard VAE        (b) Distortion-Targeted VAE

Figure 2: Samples from a standard VAE with a trainable variance versus a VAE targeted to achieve the same distortion. Both have poor sample quality despite achieving good ELBOs (Figure 1).

In this section, we highlight the challenges of applying variational autoencoders with Gaussian observation models to continuous data. We seek to train a model $p_\theta(x, z_{0:T})$ with observation $x$ and latents $z_{0:T}$. We use a hierarchical prior $p_\theta(z_{0:T}) = \prod_t p_\theta(z_t \mid z_{<t})$ and a Gaussian observation model $p_\theta(x \mid z_{0:T})$. As many practitioners may be aware and as demonstrated in Figures 1 and 2, training Gaussian observation VAEs with a trainable variance presents key challenges.

**Optimization difficulty.** As past works have noted, training variational autoencoders (VAE) is challenging (Rezende & Viola, 2018; Vahdat & Kautz, 2020; Child, 2020). It seems, however, that VAEs with trainable-variance Gaussian observation models exacerbates the issue. Even with modeling heuristics such as gradient skipping and precision-weighted merging that are intended to ease the optimization landscape (Child, 2020; Sønderby et al., 2016), we observe that optimization easily becomes unstable (Figure 1). To overcome this issue, we instead train a distortion-targeted VAE—more precisely, we use a mean-squared error reconstruction loss but tune the reconstruction coefficient to achieve the same mean-squared error as the standard VAE.

**Good ELBOs but poor sample quality.** In Figure 1, we show that both the standard VAE and distortion-targeted VAE achieve small distortion and—as we shall see demonstrate in subsequent sections—good Evidence Lower Bound (ELBO). However, this does not translate to good sample quality (Figure 2).

Since distortion-targeting appears to largely resolve the optimization issue, the primary goal of our work is to explain—and ultimately fix—why VAEs achieve good ELBOs but poor sample quality.

# 3 PERCEPTUALLY IMPORTANT INFORMATION IN VAEs

## 3.1 MEAN SAMPLING IN VAEs

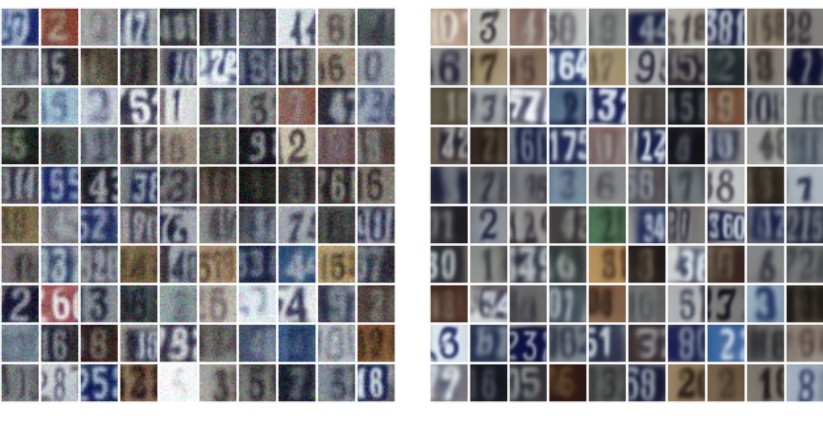

(a) Standard Sampling          (b) Mean Sampling

Figure 3: Distortion-targeted VAE with a non-negligible distortion target $\gamma = 16$, trained on SVHN. Standard sampling from $p_\theta(x)$ versus mean sampling from $\mu_\theta(Z_{0:T})$. Standard sampling with a Gaussian observation model is noisier since it essentially adds Gaussian noise to the mean samples.

A common practice for sampling from variational autoencoders involve sampling the latent code $z_{0:T}$ followed by decoding the mean of $p_\theta(x \mid z_{0:T})$, as described by

$$z_{0:T} \sim p_\theta(z_{0:T}) \tag{1}$$
$$\mu(z_{0:T}) = \mathbb{E}_{p_\theta(x \mid z_{0:T})}[x]. \tag{2}$$

In other words, letting $Z_{0:T}$ denote the random variable distributed according to $p_\theta(z_{0:T})$, practitioners often report the sampling from $\mu(Z_{0:T})$ rather than actually sampling $x \sim p_\theta(x)$—especially when the choice of the decoder makes $\mu(z_{0:T})$ easy to calculate, such as in the case when $p_\theta(x \mid z_{0:T})$ is a Gaussian observation model. The design choice to sample the mean $\mu(Z_{0:T})$ instead of the full generative process $p_\theta(x)$ is of technical importance, since the *implicitly* defined distribution $\mu(Z_{0:T})$ does not necessarily have a well-defined density. This is because $\mu(Z_{0:T})$ is not guaranteed to cover a non-zero measure in $\mathcal{X}$.

Crucially, the benefit of sampling from $\mu(Z_{0:T})$ instead of $p_\theta(x)$ comes down to how we choose to handle the information *not* encoded by $Z_{0:T}$. Since the goal of $p_\theta(x \mid z_{0:T})$ is to model the residual information in the data distribution lost during the lossy encoding via $q_\phi(z_{0:T} \mid x)$, the mean sample $\mu(Z_{0:T})$ chooses to handle this information by averaging over it. In contrast, $p_\theta(x \mid z_{0:T})$ attempts to explicitly model this information. In the case where $p_\theta(x \mid z_{0:T})$ is a Gaussian observation model, the variational autoencoder is essentially attempting to crudely model the residual information via a Gaussian distribution. In Figure 3, we compare the resulting images from sampling $p_\theta(x)$ versus $\mu(Z_{0:T})$. Whereas the former yields a noisy image, the latter averages over the noise and yields a blurry image. In cases where we are willing to accept the information loss, mean sampling offers a more visually-appealing alternative to sampling from $p_\theta(x)$. We commit to using mean sampling in all subsequent figures.

## 3.2 HOW MUCH INFORMATION ACTUALLY MATTERS PERCEPTUALLY?

In this section, we scrutinize the behavior of the mean decoding function $\mu(\cdot)$. In particular, we are interested in bounding the mutual information $I(X ; Z)$ necessary for the variational autoencoder to achieve perceptually high-quality reconstructions. Recalling that $\mu_\theta(z_{0:T}) := \mathbb{E}_{p_\theta(x \mid z_{0:T})}[x]$, we consider the objective

$$\underset{\theta, \phi}{\text{minimize}} \; \mathbb{E}_{p_{\text{data}}(x)} \left[ D_{\text{KL}}(q_\phi(z_{0:T} \mid x) \parallel p_\theta(z_{0:T})) \right] \tag{3}$$

$$\text{subject to } \mathbb{E}_{p_{\text{data}}(x)} \mathbb{E}_{q_\phi(z_{0:T} \mid x)} \|x - \mu_\theta(z_{0:T})\|^2 = \gamma. \tag{4}$$

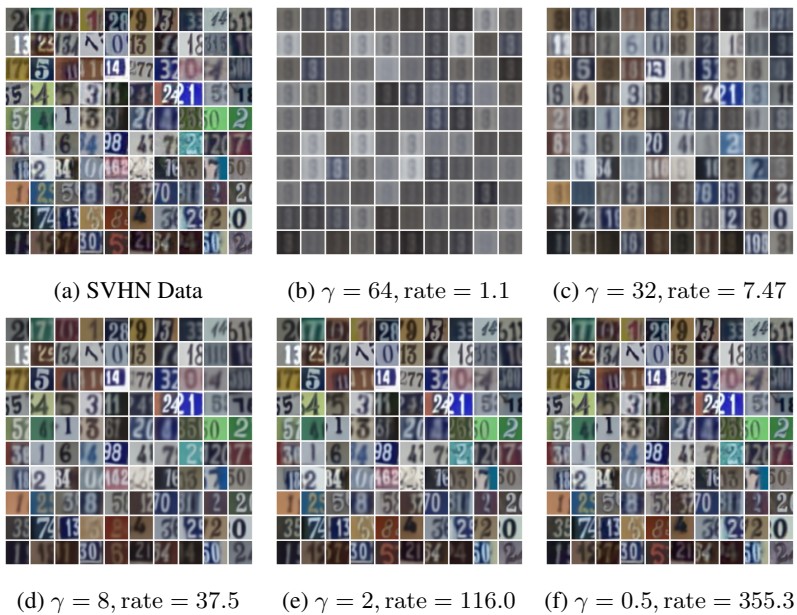

(a) SVHN Data   (b) $\gamma = 64$, rate $= 1.1$   (c) $\gamma = 32$, rate $= 7.47$

(d) $\gamma = 8$, rate $= 37.5$   (e) $\gamma = 2$, rate $= 116.0$   (f) $\gamma = 0.5$, rate $= 355.3$

Figure 4: Reconstructions from distortion-targeted VAEs for various choices of $\gamma$. Choosing $\gamma = 8$ suffices to achieve a high-quality reconstruction and requires encoding significantly less information (rate) into the latent space than $\gamma = 0.5$.

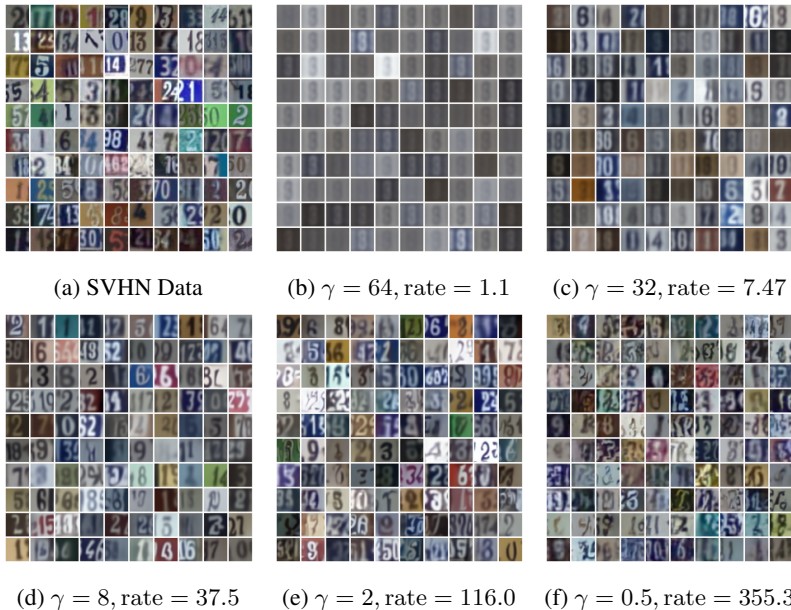

(a) SVHN Data   (b) $\gamma = 64$, rate $= 1.1$   (c) $\gamma = 32$, rate $= 7.47$

(d) $\gamma = 8$, rate $= 37.5$   (e) $\gamma = 2$, rate $= 116.0$   (f) $\gamma = 0.5$, rate $= 355.3$

Figure 5: Samples from distortion-targeted VAEs for various choices of $\gamma$. Lower rates result in easier modeling but blurrier samples. Higher rates make the modeling task much more challenging, causing poor sample quality despite good reconstructions (Figure 4)

We optimize this constrained objective using a Lagrange multiplier (Zhao et al., 2018). We note that the rate term $D_{\mathrm{KL}}(q_\phi(z_{0:T} \mid x) \parallel p_\theta(z_{0:T}))$ is an upper bound on the mutual information $I(X; Z_{0:T})$ under the distribution defined by $p_\theta(x)q_\phi(z_{0:T} \mid x)$ (Alemi et al., 2018). By varying the choice of $\gamma$, we can modulate the amount of information about $x$ that is stored in in the latent code $z_{0:T}$. Figure 4 shows the reconstructions along with the rate of the learned model. Models with high rate contain considerably more information about $x$ in the latent space and adds additional burden that the prior

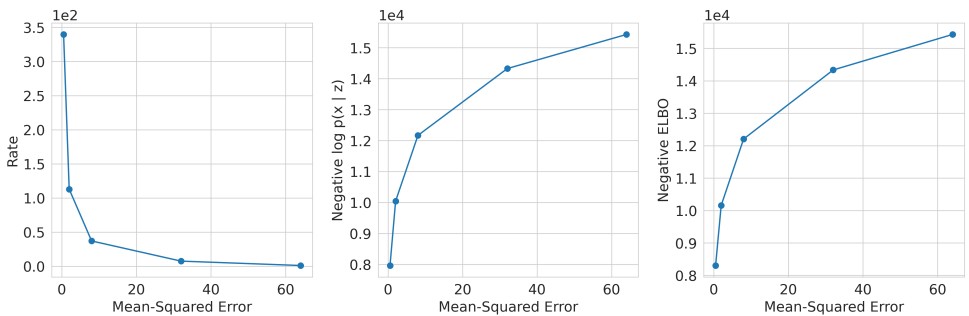

Figure 6: Rate, conditional log-likelihood $\ln p_\theta(x \mid z_{0:T})$, and ELBO as a function of the distortion achieved by various VAEs on SVHN. The ELBO favors high-rate models, which has poor sample quality (Figure 5).

$p_\theta(z_{0:T})$ needs to model correctly. In Figure 5, we show the corresponding samples generated by $\mu_\theta(Z_{0:T})$ when $Z_{0:T}$ is distributed according to $p_\theta(z_{0:T})$. Note that smaller the rate, the more distributionally-similar the reconstructions (Figure 4) are to the samples (Figure 5). Furthermore, Figure 6 shows that the ELBO objective favors high-rate models, which—when naively trained— have poor sample quality as shown in Figure 5.

## 4 PRIORITIZING THE BITS THAT MATTER

In contrast to naive training, which favors extremely small distortions and high rates, our experiments demonstrate that only a small amount of information is required to achieve high sample-quality reconstructions. Furthermore, by limiting the the amount of information encoded into the latent space, the full modeling capacity of the prior $p_\theta(z_{0:T})$ can be dedicated to the small amount of information necessary for perceptual quality, which in turn improves the sample quality of $\mu(Z_{0:T})$ when $Z_{0:T}$ is drawn from the prior $p_\theta(z_{0:T})$.

The drawback, however, is that the large amount of entropy that remains is now crudely modeled by a Gaussian observation model, causing the model to achieve poor likelihoods. To address this issue, we now train a second variational autoencoder—with parameters $(\tilde{\phi}, \tilde{\theta})$ and latent variables $\tilde{z}_{0:T}$—conditional on the mean decoding $\tilde{x} := \mu(z_{0:T})$ achieved by the first model, via the objective

$$\underset{\tilde{\theta}, \tilde{\phi}}{\text{minimize}} \; \mathbb{E}_{q_\phi(x, z_{0:T})} \left[ D_{\text{KL}}(q_{\tilde{\phi}}(\tilde{z}_{0:T} \mid x, \tilde{x}) \| p_{\tilde{\theta}}(\tilde{z}_{0:T} \mid \tilde{x})) \right] \tag{5}$$

$$\text{subject to} \; \mathbb{E}_{q_\phi(x, z_{0:T})} \mathbb{E}_{q_{\tilde{\phi}}(\tilde{z}_{0:T} \mid x, \tilde{x})} \| x - \mu_{\tilde{\theta}}(\tilde{z}_{0:T}, \tilde{x}) \|^2 = \tilde{\gamma}. \tag{6}$$

For national simplicity, we subsumed $p_{\text{data}}$ by defining $q_\phi(x, z_{0:T}) := p_{\text{data}}(x) q_\phi(z_{0:T} \mid x)$.

Our two key insights are as follows. First, the reconstructions from the first variational autoencoder augments the initial dataset by creating the paired data of the form $(x, \tilde{x})$. This paired data subsequently defines a new conditional density estimation problem that seeks to predict $x$ based on the learned reconstructions. Second, the secondary variational autoencoder can trivially recover the initial distortion $\tilde{\gamma} = \gamma$ by setting the decoder as $\mu_{\tilde{\theta}}(\tilde{z}_{0:T}, \tilde{x}) = \tilde{x}$. And any further improvement is thus bounded by $\gamma$. Furthermore, since the initial distortion $\gamma$ was chosen to achieve perceptually-acceptable reconstructions, the secondary variational autoencoder is thus restricted to making small perturbations to the existing decoding $\tilde{x}$.

Taken together, we argue that since the perceptible bits are already accounted for by the primary variational autoencoder, the secondary variational autoencoder is thus restricted to modeling the remaining imperceptible bits. As such, we hypothesize that not only will the secondary variational autoencoder improve upon the initial ELBO by replacing the crude Gaussian observation model with a more expressive conditional VAE observation model, it will also do so with minimal impact to the sample quality achieved by the primary variational autoencoder. Figure 7 confirms this hypothesis, and Figure 8 further demonstrates that the secondary variational autoencoder significantly improves the initial ELBO without sacrificing sample quality.

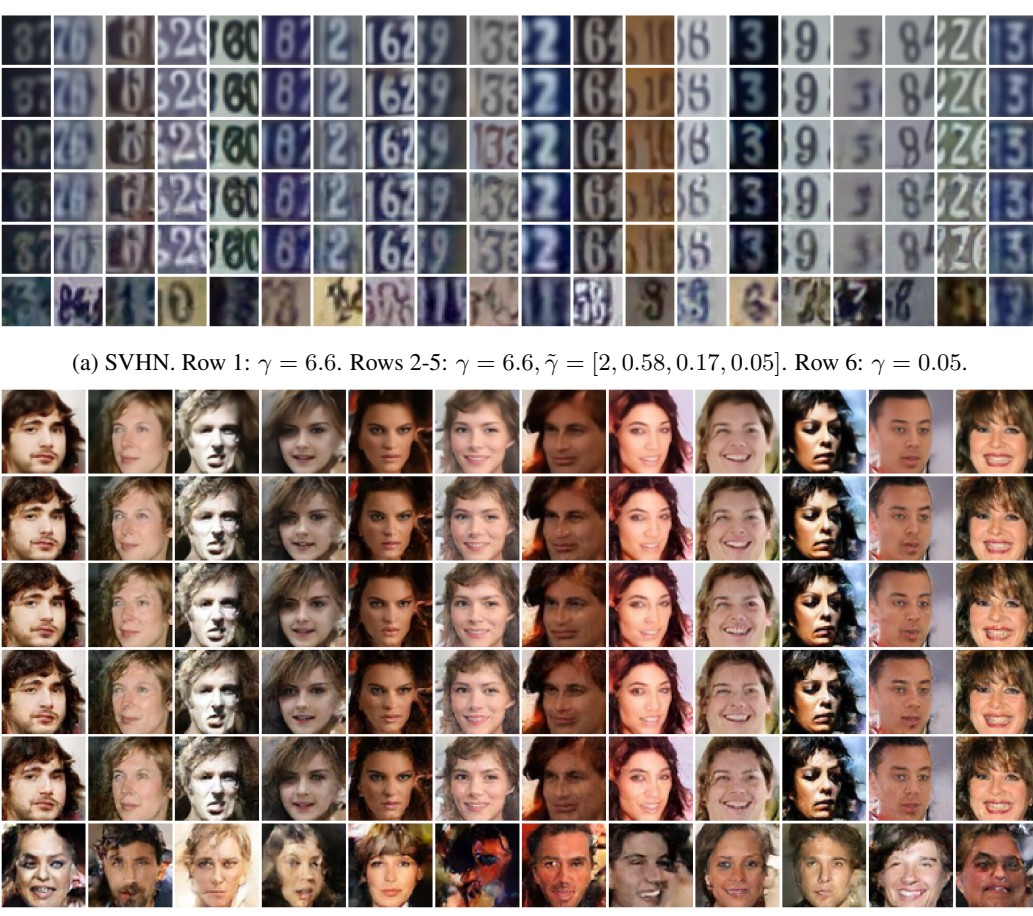

(a) SVHN. Row 1: $\gamma = 6.6$. Rows 2-5: $\gamma = 6.6, \tilde{\gamma} = [2, 0.58, 0.17, 0.05]$. Row 6: $\gamma = 0.05$.

(b) CelebA. Row 1: $\gamma = 13.8$. Rows 2-5: $\gamma = 13.8, \tilde{\gamma} = [3, 2, 0.58, 0.3]$. Row 6: $\gamma = 0.3$.

Figure 7: Samples from two-stage training for SVHN and CelebA. The first row shows samples from the primary VAE. The next four rows show samples from secondary VAEs with various choices of $\tilde{\gamma}$. The secondary VAE makes near-imperceptible visual changes to the primary VAE's samples. The last row shows samples from a VAE that directly targets a small distortion. In contrast to two-stage training, directly targeting a small distortion yields poorer sample quality.

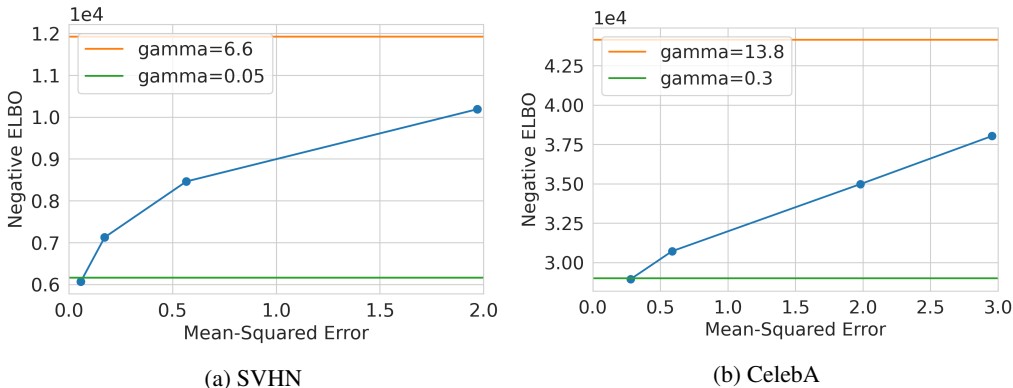

(a) SVHN

(b) CelebA

Figure 8: Test set ELBO from two-stage training as a function of the final distortion achieved. ELBOs for single-stage training are shown as horizontal lines for comparison. For the same final distortion target, two-stage training achieves similar ELBO as single-stage training while also having better sample quality (Figure 7).

## 5   NOT ALL MEAN-SQUARED ERRORS ARE CREATED EQUAL

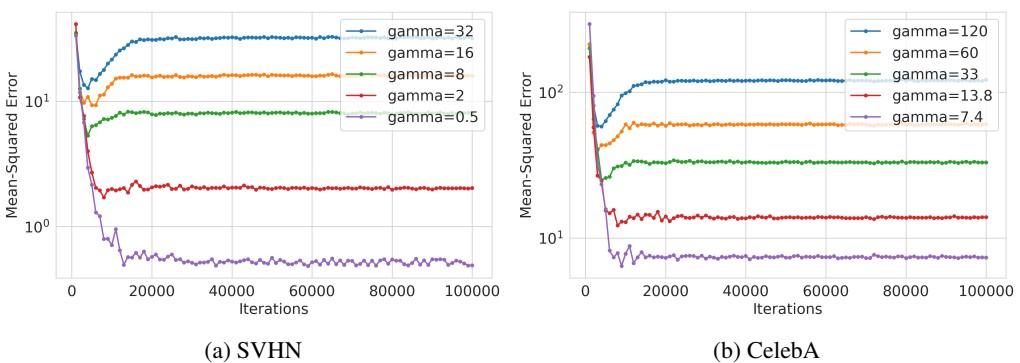

(a) SVHN                                   (b) CelebA

Figure 9: Mean-Squared Error over the first 100000 iterations. All models quickly converge to the targeted distortion values.

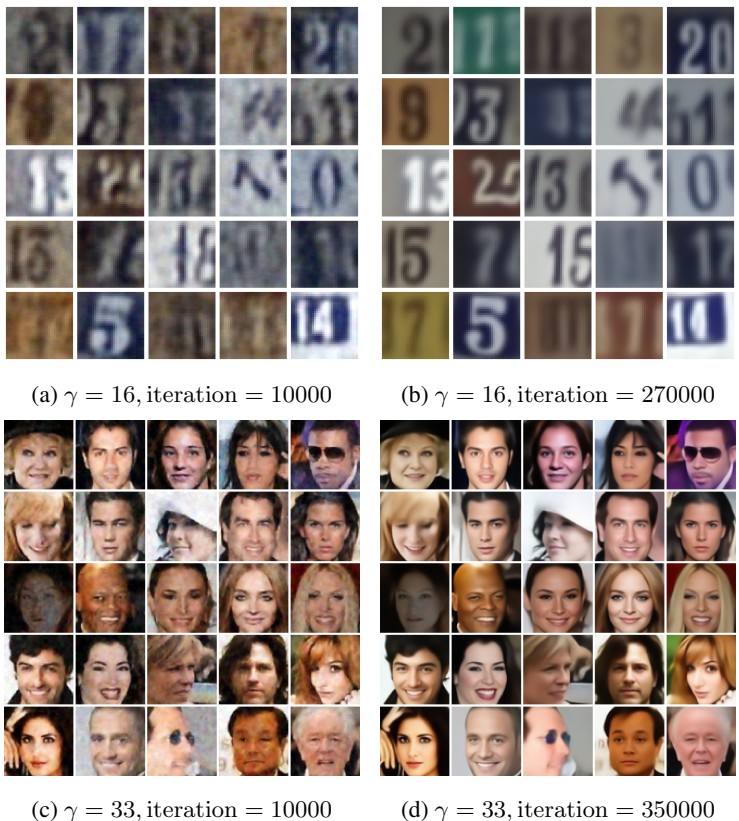

(a) $\gamma = 16$, iteration $= 10000$            (b) $\gamma = 16$, iteration $= 270000$

(c) $\gamma = 33$, iteration $= 10000$            (d) $\gamma = 33$, iteration $= 350000$

Figure 10: Reconstructions for SVHN and CelebA as a function of the distortion target $\gamma$ and training iterations. Despite achieving the same distortion (Figure 9), earlier iterations capture higher-frequency information.

While the primary focus of our work is on prioritizing the modeling of visually perceptible information, we also observe over the course of training a notable characteristic of how mean-squared error relates to the visual quality of the reconstructions. Figure 9 shows characteristic distortion over time curves when training the primary variational autoencoder on SVHN and CelebA for various choices of $\gamma$. All models quickly achieve the targeted distortion value; the remainder of the training thus focuses on minimizing the rate while keeping the distortion constant.

A natural consideration, then, is how the reconstruction differs visually near the beginning versus the end of training despite having the exact same mean-squared error. In Figure 10, we see that the reconstructions have much greater global consistency near the end of training than at the beginning of training. We interpret these results as indicating that the models rapidly achieve the desired distortion targets by encoding high-frequency information, but gradually shifts to using lower-frequency information (and thus achieving lower rate) over the course of training.

To draw an analogy to principal components analysis, for the bottom $k$ eigenvectors to achieve the same reconstruction cost as the top-$k'$ eigenvectors, it tends to be the case that $k \gg k'$ for natural images. Similarly, while there are many encoding schemes that can achieve the reconstruction cost, they may differ greatly in the amount of information stored in the latent space. Masking out part of the image, for example, is an encoding scheme that still preserves significant amount of fine-grained texture information in the image and will likely result in a much higher rate. As a result, the constrained objective favors blurrier, but globally-consistent reconstructions over (locally) sharper but globally-inconsistent reconstructions.

## 6 RELATED WORK

Given the extensive literature on deep generative models, we highlight several works most relevant to our paper. Theis et al. (2015) pointed out that good likelihoods do not imply good sample quality, and showed that it is possible to construct pathological distributions that exhibit such behavior (by mixing real with data with noise data). However, such a pathology does not reflect the failure mode that variational autoencoders exhibit in practice, which is the focus of our work. Meng et al. (2021); Dai & Wipf (2019) both proposed two-stage training processes, but with different underlying motivations. Meng et al. (2021) proposed to train a primary model on a noised version of the original dataset to ease optimization in autoregressive models, whereas our two-stage training procedure explicitly characterizes the relation between likelihood and sample quality. Dai & Wipf (2019) motivated their two-stage training process to handle data manifolds with intrinsic dimensionality lower than the dimensionality of the ambient space $\mathcal{X}$. In contrast, both our primary and secondary models are deep hierarchical VAEs whose number of latent variables exceed the dimensionality of $\mathcal{X}$. Menick & Kalchbrenner (2018) also decomposes the original image modeling problem into bits that are more versus less significant. However, the Subscale Pixel Network prescribes a hand-designed encoding scheme, whereas we allow the model to learn an encoding scheme based on what minimizes rate most for a particular distortion target. Higgins et al. (2016) bears resemblance to our primary model, but differ in how the distortion is targeted; whereas $\beta$-VAE tunes the coefficients of reconstruction and rate terms, we directly target mean-squared error value, which is more interpretable. Finally, both VDVAE and NVAE (Child, 2020; Vahdat & Kautz, 2020) highlight the optimization challenges in VAE training; our work builds on of these works by further addressing the challenge of building VAEs that simultaneously achieve good sample quality and good likelihoods.

## 7 CONCLUSION

In this work, we critically examined why variational autoencoders tend to exhibit good likelihoods but poor sample quality for image generative modeling. Our experiments demonstrate that only a small amount of information in natural image distributions is pertinent to perceptual sample quality. However, conventional ELBO optimization does not distinguish the bits that matter perceptually from the bits that do not. And since the imperceptible bits dominate the ELBO signal, the model does not dedicate enough modeling capacity to the perceptually relevant information, thus causing poor sample quality. In addition to demonstrating this phenomenon, we also propose a simple two-stage training procedure that prioritizes the modeling the perceptible information. By doing so, we can reliably train VAEs with good sample quality while still achieving ELBOs comparable to conventionally-trained VAEs. While our work focuses on variational autoencoders, we believe that this general phenomenon of imperceptible information dominating the likelihood signal is relevant to other likelihood-based models for image modeling. Our work demonstrates not all bits are created equal, and we encourage researchers and practitioners alike to explicitly prioritize the bits that "matter" in the design of deep generative models.

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

# A   ARCHITECTURE

## A.1   PRIMARY VAE

Our model makes use of the VDVAE architecture (Child, 2020) in conjunction with precision-weighted merging (Sønderby et al., 2016). For SVHN, we use a model with 48 layers of stochasticity, consisting of two $1 \times 1$ resolution layers, four $4 \times 4$, ten $8 \times 8$, sixteen $16 \times 16$, and sixteen $32 \times 32$ layers. For CelebA, we simply added an additional eight $64 \times 64$ resolution layers. Each layer of latent variables at any $n \times n$ resolution has 16 channels. The hidden dimensionality of the residual network is fixed at 128 channels, and all bottleneck residual blocks uses a 32 channel bottleneck.

## A.2   SECONDARY VAE

The secondary VAE has the extract same depth-structure as the primary VAE and simply conditions on the additional image $\tilde{x}$ generated by the primary VAE. For the inference model, we simply concatenated $[x, \tilde{x}]$ as the input. For the generative model, we use a U-Net structure to inject information about $\tilde{x}$ at all resolutions in the generative process.

# B   TRAINING INFORMATION

## B.1   DEQUANTIZATION

We dequantize both SVHN and CelebA by adding uniform noise in the interval $(0, 1)$ to the original $[0, 255]$ pixel intensities. We only use the dequantized images when computing $\ln p_\theta(x_{\text{dequantized}} \mid z_{0:T})$. The inference model is always given the clean image. This practice of feeding the clean image into the inference model still admits a valid ELBO.[1]

## B.2   LAGRANGIAN

Letting $R(\phi, \theta)$ and $D(\phi, \theta)$ denote the rate and distortion (as measured by mean-squared error) respectively, we can follow Zhao et al. (2018) and perform $\gamma$ distortion-targeting by jointly optimizing the following two objectives with gradient descent.

$$\underset{\phi,\theta}{\text{minimize}}\, R(\phi, \theta) + D(\phi, \theta) + \lambda \cdot (D(\phi, \theta) - \gamma) \tag{7}$$

$$\underset{\lambda}{\text{maximize}}\, \lambda(D(\phi, \theta) - \gamma). \tag{8}$$

In practice, we instead perform

$$\underset{\phi,\theta}{\text{minimize}}(1 - \lambda) \cdot R(\phi, \theta) + \lambda \cdot D(\phi, \theta) \tag{9}$$

$$\underset{\lambda}{\text{maximize}}\, \lambda(D(\phi, \theta) - \gamma). \tag{10}$$

The general intuition still holds: if $D > \gamma$, then Equation (10) is maximized by increasing $\lambda$, which in turn encourages Equation (9) to reduce $D$. We restrict $\lambda \in [0.0001, 0.9999]$ via projected gradient descent and initialize at $\lambda = 0.9999$ to mimic KL annealing Sønderby et al. (2016).

## B.3   OPTIMIZATION

We use the Adam optimizer (Kingma & Ba, 2014). We use a learning rate of $2 \times 10^{-4}$ for both Equations (9) and (10). However, we use $(\beta_1, \beta_2) = (0.9, 0.999)$ for Equation (9) but $(\beta_1, \beta_2) = (0.0, 0.999)$ for Equation (10).

---

[1]see http://ruishu.io/2018/03/19/bernoulli-vae/ for the lower bound interpretation.

