# OpenReview forum: "Mind Your Bits and Errors: Prioritizing the Bits that Matter in Variational Autoencoders"
_ICLR.cc/2022/Conference — ICLR 2022 Submitted_

### Official Review · Reviewer_KKon · 2021-11-01

**Correctness:** 2
**Technical Novelty And Significance:** 3
**Empirical Novelty And Significance:** 4
**Recommendation:** 5
**Confidence:** 4

**Main Review:**

The topic of exploring over-parameterized VAEs is really important. The authors try to analyze the reasons of model’s poor performance. Over-parameterized hierarchical VAEs are able to achieve perfect reconstruction on the training set but become extremely vulnerable to overfitting. So it is important to understand how one can mitigate this issue. Authors suggest an interesting form of regularization and show that it improves the quality of generation at the expense of ELBO value. To deal with the latter they build secondary VAE upon the primary and show that now both good reconstruction and ELBO are achieved.
Although I liked the general concept I still recommend rejection of the paper in its current form due to several reasons:
The authors claim that the reason of poor performance is “model being overwhelmed by the vast volume of visually-imperceptible information”. The claim itself may be right (although I disagree with it – see below) but is never shown in experiments. All that is observed/reported can be explained by more simple and clear reason: overfitting on training data. It may happen in over-parameterized VAE that are perfect in reconstructing training objects. They achieve arbitrary low reconstruction error and very high ELBO but have a mismatch between marginalized encoder $ \int q(z|x) p_{data}(x) dx$ and prior $p(z)$. The reason of this mismatch is the domination of reconstruction term that can be made very large in case of (almost) perfect reconstruction since std in MSE loss goes to zero. In this case the mismatch is not penalized and hence further regularization is necessary. This is exactly what authors do but I do not think it is related to overwhelming of model. I have not found any experiments that confirm that VAE is concentrating on visually imperceptible information at the expense of visually important details. So I would say that the main claim remains unproven if not wrong.
Does your model suffer from mode-collapse (that is when you posterior q(z|x) collapses to prior p(z))? This problem was reported for over-parameterized VAE and it seems that minization of the rate given reconstruction error fixed can make the situation even worse increasing the number of «dead latents». It would nice to see the comparison between their quantity in initial VAE and in the proposed two-stage model.
The paper looks like it was written in a rush. There are many figures and few experiments. The phenomena and suggested modifications of the model is important and should be studied in more details. There is enough space for that in the paper. In particular I think it is important to see:
The comparison with less over-parameterized VAE that directly optimizes ELBO and is able to achieve the same value as the primary VAE that minimizes rate.
The relation between the values of gamma and values of beta in beta-VAE. What should be the value of beta in order to achieve the same rate and reconstruction as is achieved by primary VAE with given value of gamma?
Current scheme is two-stage. Is it possible to derive single model that could be trained end-to-end and where losses from primary and secondary VAE are combined? That would be a nice regularization method for over-parameterized VAE.
The value of gamma is a crucial hyperparameter. It is important to see the strategies for its selection for different datasets.
How the rate changes in the last experiment? Is there any significant change between iterations 10000 and 270000/350000?




**Summary Of The Paper:**

The paper investigates the possible reasons for poor performance of over-parameterized VAE that are able to achieve good train ELBO but fail to generate realistically-looking objects. The authors formulate a hypothesis and try to prove it experimentally deriving a novel two-stage scheme for VAE that provides good ELBO values and generates high-quality images. In particular they suggest to minimize rate (that is KL divergence between approximate posterior and prior over the latents) given the reconstruction error does not exceed predefined threshold gamma. This serves as a kind of regularizer that prevents overfitting and allows for better generation.

**Summary Of The Review:**

Overall I find the paper interesting but in its current form it is more a workshop paper. Many important experiments are missing. The main claim is not justified and may be wrong. I encourage the authors to improve the paper, to better study the reasons that cause poor generation and to resubmit.

---

> ### Author Response · Authors · 2021-11-23
> **response**
>
> Thank you for thoughtful review! We would like to address your concerns with this paper and hope to continue improving on our paper so that the main message is clear and convincing.
>
> ### Regarding the “mismatch” hypothesis
>
> Thank you for bringing forth this potential alternative hypothesis. To our understanding, your suggested hypothesis would not explain how to construct a high-rate low-reconstruction-error model with good sample quality. In contrast, our hypothesis predicts that a two-stage model that first prioritizes the perceptually relevant information and then perceptually less-relevant information can in fact construct a high-rate low-reconstruction-error model with good sample quality---as demonstrated by Figure 7 and 8. The fact that, (1) naively training a high-rate low-reconstruction-error model leads to poor sample quality but, (2) two-stage guided training yields high-rate low-reconstruction-error model with good sample quality, suggests the naively-trained model is overwhelmed. We thus believe that our hypothesis has better predictive power and a better fit to the experimental results shown in the paper. If the reviewer has any other counterhypothesis or explanations in mind, and wish to suggest experimental designs to address such confounders, we are excited to conduct/discuss them!
>
> ### Posterior collapse
>
> We believe our model cannot experience posterior collapse because of the specific reconstruction values we targeted. Gaussian VAEs that experience posterior collapse have reconstruction error no better than the mean-squared error to the average image. So long as the targeted reconstruction value is lower than that and is successfully achieved, posterior collapse (where q(z|x) matches p(z)) is impossible.
>
> ### Selection of gamma
>
> Since the selection of gamma is ultimately a matter of perceptual quality, we selected gamma in the paper simply via visual inspection. As suggested by another reviewer, we will improve the experimental section of our paper to be more systematic by including sample quality metrics and a more programmatically reproducible gamma-selection process.
>
> ### Derivation of a single end-to-end model
>
> Since the goal of our paper is to demonstrate the hypothesis of the “model being overwhelmed by the vast volume of visually-imperceptible information”, the design of a single end-to-end model is currently outside the scope of our paper. However, we believe the insights from our paper will be helpful in designing such a model in the future and are actively working on it.
>
> ### Is there a significance rate change between iterations 10000 and  270000/350000.
>
> Yes. For example, in the CelebA experiment with gamma=33, we observed a rate of ~1000 at iteration 10k and a rate of ~250 at iteration 350k. A more careful experiment, however, would require that we freeze the encoder at iteration 10k and train the decoder for more steps to rule out underfitting concerns.

---

> > ### Comment · Reviewer_KKon · 2021-11-30
> > **Thanks for the response - still keep my score**
> >
> > Dear authors,
> >
> > Thanks for a response! My main point for criticism still holds. You claim that a naively-trained model is overwhelmed while I say that it does exactly what it should from training objective point of view i.e. provides perfect reconstructions and high ELBO values. This happens because over-parameterized model can easily fit training data that covers only a fraction in latent space (it becomes more profitable to derive q(z|x) with tiny variances) hence random samples from prior are not covered by any q(z|x) and lead to poor generation. This is all about overfitting. Your idea of maximizing rate seems to be a good regularizer and it really helps. However your interpretation of what is going on is doubtful. I am not claiming that you are wrong – I am just stating that I do not see the arguments in favor of your claim in the paper. All what is described there can be explained in a more simple way – just overfitting of over-parameterized model. So I am afraid that I can recommend the paper as a workshop paper but not for acceptance to conference.
> >
> > I also hope that after reviewing period is over I will be able to find the authors and to get in touch with them to discuss the observed effect in more details. In particular I am interested why your two-stage model achieves the same ELBO as naively-trained model (see fig. 8)? Is it typical situation? If so does it mean both VAEs achieve global minima of train loss but completely different ones?

---

> > > ### Author Response · Authors · 2021-12-01
> > > **response**
> > >
> > > Thanks for the response! And we look forward to continuing this discussion beyond the conference review. I think it is possible that we may be proposing related hypotheses of the same underlying phenomenon at different levels of abstraction (akin to macroscopic vs microscopic theories). I look forward to having a more in-depth discussion to see where we can land on the same page regarding how VAEs behave!
> > >
> > > As for why the two-stage model achieves similar ELBO, it is worth noting that the first-stage model encodes significantly less rate (an order of magnitude) than the naively-trained model and second-stage model. Even if the second-stage model were to completely ignore the first-stage model, it would only mean "wasting" a relatively small amount of rate. Since the second-stage model is taking advantage of the first-stage model's output, and perhaps combined with added optimization ease (appealing to Meng et al's distribution smoothing argument), any "rate-waste" caused by the two-stage approach may ultimately be negligible. Whether or not these models qualify as achieving global minima, we can attest that these models achieve similar training losses but with very different perceptual sampling behavior, and that this appears to be reproducible phenomenon in our SVHN/CelebA experiments.

---

### Official Review · Reviewer_LBJj · 2021-11-01

**Correctness:** 2
**Technical Novelty And Significance:** 3
**Empirical Novelty And Significance:** 2
**Recommendation:** 3
**Confidence:** 4

**Main Review:**

The proposed method is interesting. It resembles the Laplace pyramid approach to image modelling (high-frequency information conditioned on lower-frequency information), but in the proposed method we model "imperceptible" information conditioned on the "perceptible" information, motivating it with the fact that it's the "imperceptible" information that contributes most to the likelihood.

I have a few issues with the paper, however. First, evaluation of the proposed method is not rigorous enough. Experiments do show that the model improves on the single-stage baseline in terms of ELBO-at-given-rate and sample-quality-at-given-rate, but 1) no quantitative perceptual quality metrics (e.g. FID) are used 2) no third-party baselines are used, especially other VAEs that aim to improve perceptual quality of samples (two-stage VAE by Dai & Kipf (2019), beta-VAE, VAEs with perceptual loss, etc.). In addition, the computational/parameter complexity is not discussed: we have to train *two* models instead of one in the proposed method, which is more expensive and increases the number of parameters when compared to the single-stage baseline. This should be accounted for in comparisons.

Third, while I do commend the authors for building a narrative to motivate the proposed method, I've struggled to relate much of the content to prior work, and found some of it not to be sufficiently novel. A few examples follow.

The notion of the rate-distortion (RD) trade-off/curve in relation to VAE learning (Alemi et al., 2018) is not properly introduced, even though what eqs. (3)-(4) achieve is they select a particular point on the RD Pareto front. Doing so was also proposed by Alemi et al. in Section 2, hence the novelty of this should be discussed.

The finding that both low-rate and high-rate models result in poor samples is not surprising to me: in the former case the posterior doesn't deviate enough from the prior to store any information about the input, hence the decoder has to output an "average" blurry image; in the latter case the posterior is *too* different from the prior, so samples from the prior are likely out of the decoder's distribution. The finding that ELBO favours high-rate models is also not ironclad: standard ELBO can be seen as an instance of beta-weighted ELBO with beta=1 (Higgins et al., 2017), and using beta > 1 can make ELBO favour lower-rate/higher-distortion solutions.

Finally, Section 2 and Section 3.1 are of questionable value. Section 2 ends with "distortion-targeting appears to largely resolve the optimization issue", which is much too broad a conclusion for the results presented. Figure 3 shows a that samples from a Gaussian are "noisier" than the mean of that Gaussian, which is hardly surprising.

Minor points:
- "Rate" and "distortion" are used throughout the write-up, but neither is at any point defined precisely.
- Consider discussing [1], a more theoretical treatment of the relationship between the RD curve and perceptual quality.
- Avoid figures immediately following a section title: makes for difficult reading.
- "MSE" and "distortion" are used interchangeably: I'd stick to one.
- Typos: "for _national_ simplicity"; in sec. 7: "the modeling _ the perceptible information".

References:
- [1]: https://arxiv.org/abs/1901.07821

**Summary Of The Paper:**

Authors explore a disconnect between likelihood and perceptual quality in likelihood-based generative models --- a well-known phenomenon (Theis et al., 2015). Authors take the rate-distortion perspective of VAEs due to Alemi et al. (2018), and demonstrate that high-rate models (i.e. models that store more information about the input in the latent variables) tend to produce samples with poor perceptual quality. This is problematic, as authors also show that the standard ELBO training objective prefers such higher-rate models. In attempt to alleviate this authors propose a two-stage method, where one VAE is trained to be low-rate (i.e. to have high-quality samples), and a second VAE is trained to be high-rate and model the distribution of *imperceptible information* conditioned on the output from the first VAE. Authors run experiments to demonstrate that the proposed method produced samples with high-perceptual quality, while also having high test likelihoods.

**Summary Of The Review:**

While I do find the proposed method interesting and thought-provoking, I don't think it's evaluated with sufficient rigour. I also find the rest of the story in the paper to not be sufficiently novel or well-connected to prior work. In the end, I consider the paper to be below the bar for acceptance.

---

> ### Author Response · Authors · 2021-11-23
> **response**
>
> Thank you for thoughtful review! We would like to address your concerns with this paper and hope to continue improving on our paper so that the main message is clear and convincing.
>
> ### Relation to existing work on rate-distortion trade-off
>
> We wish to note that the Eq (3-4) is not the claim to novelty in the paper. It is, after all, simply a tool for targeting a particular rate-distortion value. The novelty in our work is in showing that:
> 1. We do not actually need to model a lot of information to achieve perceptually satisfactory samples (Fig 4)
> 2. When you do try to model a lot of information at once, there is no reason for the prior distribution model to prioritize the perceptually important information (Fig 5).
>
> Collectively, we interpret these findings to mean that it is not that high-rate is inherently bad, but simply that naively training a high-rate model and hoping for it to be good at sample quality is an unlikely endeavor---akin to finding a needle in a haystack. Finally, the goal of the two-stage training procedure is thus a demonstration that we can indeed guide the training of high-rate models to achieve both good ELBOs and good sample quality, thereby finding the needle in the haystack. To our knowledge, no existing work specifically highlighted and experimentally demonstrated this line of reasoning.
>
> ### Lack of Baselines
>
> The primary goal of our work is to experimentally demonstrate the above characterization of VAEs. Whether or not other approaches also improve perceptual quality is, we argue, orthogonal to the whether our characterization is valid. We acknowledge, however, the lack of quantitative metrics in our paper; we did not use these metrics since we did not observe contemporary VAE papers (VDVAE, NVAE) to use such metrics and found that the sample quality differences shown in our experiments were large enough for visual inspection to suffice (Fig 7). Nevertheless, we recognize the value in providing quantitative metrics and will implement them in a future iteration of our work.
>
> ### Purpose of Section 2 and 3.1
>
> The goal of Section 2 and 3.1 are simply to provide a self-contained narrative and are not claims to novelty. We agree, however, that we overstepped by claiming that “​​distortion-targeting appears to largely resolve the optimization issue”. We did not mean that as a claim to novelty, but simply that fixing optimization issues (such as via gradient-skipping in VDVAE) is outside the scope of our paper. We will reword it accordingly.

---

> > ### Comment · Reviewer_LBJj · 2021-11-27
> > **Thank you; unfortunately, still sceptical**
> >
> > I thank the authors for their response. After reading the response and other reviews my assessment has not changed significantly.
> >
> > I do find the relationship between the RD curve and perceptual quality to be intriguing, and I still think the idea of the two-stage VAE is interesting as a proof-of-concept of a method that biases a likelihood-based model towards samples with high perceptual quality.
> >
> > Unfortunately, I also think the main message of the paper is muddled by weakly related figures and discussion. Even though the intended message has been somewhat clarified by the authors in their response, it is still difficult to parse from the manuscript itself in its current form. The relationship to prior work is not sufficiently precise, especially when it comes to [1-3].
> >
> > Given that the authors consider the two-stage VAE itself to *not* be the main contribution (this way explaining the lack of baselines), and rather a tool for testing their theory, I think it puts a higher bar on being focused and precise when defining the theory, being careful when relating the theory to our current understanding as expressed in prior work, as well as making sure the experiments are run under sufficiently varied conditions to provide rock-solid empirical evidence for the theory.
> >
> > I encourage the authors to take this feedback and the feedback from other reviewers into account in the next iteration of the paper, which I am looking forward to reading.
> >
> > *References:*
> > - [1]: https://arxiv.org/abs/1711.00464
> > - [2]: https://arxiv.org/abs/1901.07821
> > - [3]: https://arxiv.org/abs/1511.01844

---

### Official Review · Reviewer_7Pio · 2021-11-02

**Correctness:** 4
**Technical Novelty And Significance:** 2
**Empirical Novelty And Significance:** 2
**Recommendation:** 5
**Confidence:** 4

**Main Review:**

This paper is well written and easy to follow, and I think that further attempting to understand how likelihoods and sample quality are related is an important research direction. That being said, I do have two main problems with this paper:

1. The work of Alemi et al. [1] is merely cited in passing, and I actually believe that the objective in the reviewed paper is not really novel as compared to this work. In other words, the presented relationship between likelihoods and sample quality seems to be just a rephrasing of the rate/distortion decomposition of [1]. I would be happy to be corrected about this point if I misunderstood anything, but I strongly believe that at the very least a careful discussion about [1] and how the current work differs from it is warranted.

2. Given the previous point, I think the novel part of this paper is actually just the way in which they train VAEs in two steps. However, I don't think there is strong motivation for this: It feels like the main point of the paper is that likelihood is not important, and the only purpose of the second VAE is achieving good likelihoods. In other words, the authors spend a lot of time trying to convince the reader that likelihoods are dominated by high-frequency patters that are imperceptible to humans, and that low-likelihood-achieving VAEs are enough; only to then suggest adding a second VAE to "fix" the likelihood problem.

Minor things:

-The paper is using the ICLR 2021 format, not ICLR 2022.

-The sentence "the application of VAEs to image modeling has been lukewarm at best" is a pretty strong statement.

[1] Fixing a Broken ELBO, Alemi et al.

========================================================================================================

UPDATE 1 AFTER REBITTAL

========================================================================================================

I have read the author's response, and while I agree that this paper more closely looks at sample quality than Alemi et al., I still believe that a much more careful discussion is needed before the paper is accepted. I will thus keep my current score.

**Summary Of The Paper:**

This paper studies the relationship between likelihood and sample quality in Gaussian VAEs. The authors argue that high likelihoods are obtained through high frequency signals which are not perceptible to humans, and that low-likelihood models can actually produce good samples. The authors support this by maximizing a varational upper bound to the mutual information between latents and osbservables, with the added constraint that reconstruction errors should have a given magnitude. Finally, the authors propose a two-step procedure; where a VAE is first trained attempting to achieve reasonable samples even if it does not have good accuracy, and then a second (conditional, on the reconstructions from the first VAE) VAE tries to learn the patterns that will give large likelihoods.

**Summary Of The Review:**

While this paper studies an interesting problem in a well-presented way, I have doubts about both the novelty and the motivation of the proposed approach.

---

> ### Author Response · Authors · 2021-11-23
> **response**
>
> Thank you for thoughtful review! We would like to address your concerns with this paper and hope to continue improving on our paper so that the main message is clear and convincing.
>
> ### Relation to Alemi et al
>
> Alemi et al characterizes the trade-off between rate and distortion. Our goal is to extend this characterization to a third dimension: sample quality. We wish to note that Eq (3-4) is not the claim to novelty in the paper. It is, after all, simply a tool for targeting a particular rate-distortion value. The novelty in our work is in showing that:
> 1. We do not actually need to model a lot of information to achieve perceptually satisfactory samples (Fig 4)
> 2. When you do try to model a lot of information at once, there is no reason for the prior distribution model to prioritize the perceptually important information (Fig 5).
>
> Collectively, we interpret these findings to mean that it is not that high-rate is inherently bad, but simply that naively training a high-rate model and hoping for it to be good at sample quality is an unlikely endeavor---akin to finding a needle in a haystack.
>
> ### Purpose of the Two-stage VAE
>
> The goal of the two-stage training procedure is to demonstrate that we can indeed guide the training of high-rate models to achieve both good ELBOs and good sample quality, thereby finding the needle in the haystack. To our knowledge, no existing work specifically highlighted and experimentally demonstrated this line of reasoning. The two-stage VAE thus primarily serves as an experimental demonstration of our “needle in the haystack” hypothesis and we hope that the reviewer views it through this lens. We believe that the core message of our paper is important for deepening our understanding of sample quality in VAEs and will lead to the development of better generative models in the future.

---

### Official Review · Reviewer_sh3z · 2021-11-02

**Correctness:** 2
**Technical Novelty And Significance:** 2
**Empirical Novelty And Significance:** 2
**Recommendation:** 3
**Confidence:** 4

**Main Review:**

Strength:

The rate-distortion and related analysis has been a classic topic in understanding VAE and ELBO. This paper adds some interesting perspectives to this topic, especially from the angle of perception. This then motivated a two-stage training method that shows bits of information can be prioritized to capture more perceptible information during VAE training.

Some results are intriguing and surprising, such as the speculation that higher-frequency information is being captured during earlier iteration of VAE training (Fig 10)


Weakness:


The main aspect that needs to be clarified about this paper is its fundamental contribution. Neither the rate-distortion analysis nor the two-stage VAE is a new concept. In [1], a dedicated rate-distortion analysis has been made to show the competition and balance between these two terms in ELBO. It has been shown that a low distortion will lead to a good reconstruction, but limited generation / sampling (which is referred to as the auto-encoding limit). The findings presented from the current study seems to add little new insights to this. In general, the balance between these two terms have been widely studied, and it is not clear how much new insights this study is adding to the existing body of literature.The paper should also add better discussion of [1], which is only mentioned in passing in the current manuscript.


The presented two-stage training of VAE is also similar to existing works. In [2], for instance, a teacher-student scheme is used to prioritize the learning of the main factors of variations in the latent space (similar to prioritize the learning of the "perceptible information" in this work), while additional bits are then used to improve the image generation quality. It is true the teacher-student model is different from a two-stage model as proposed, and the motivation was posed differently -- but the fundamental contribution and insights are not clear to me given these existing works. [2] was also not discussed in this paper

Interestingly, in this paper, the authors shows that the second stage VAE decreases ELBO loss without changing perceptible quality change of the generated images. In [2], the point of the additional bits was to improve the quality of the generated images. Intuitively, one would also expect that adding more bits will allow adding more details to the images. If the second-stage VAE does not improve the perception of the image, then what is the benefit of the second-stage VAE? Just to decrease the ELBO loss? But for what actual benefits?



Experiment wise, this paper did not have a systematic experimental section. Results were presented to support statements, without a systematic description of the experimental settings. There was also no consideration or comparison to related works, further obscuring the contribution of the work compared to the state of the arts.

[1] Alexander Alemi, Ben Poole, Ian Fischer, Joshua Dillon, Rif A Saurous, and Kevin Murphy. Fixing
a broken elbo. In International Conference on Machine Learning, pp. 159–168. PMLR, 2018.

[2] Jose Lezama, Overcoming the disentanglment vs reconstruction trade-off via Jacobian supervision, ICLR, 2019

**Summary Of The Paper:**

his paper presents a rate-distortion analysis about VAE to examine why good reconstruction may not be associated with good sampling/generation. Based on the analysis, a two-stage type of training strategies that prioritizes the modeling of the "perceptible information". Analyses and presented training strategies were supported by experimental results on benchmark data such as SVHN and CelebA.



**Summary Of The Review:**

This paper discusses an interesting and long-standing topic with a perspective somewhat different from existing ones (i.e., whether information is perceptible), with statement supported by a two-stage VAE scheme. The contribution, both in insights or methods, however appears limited in terms of adding to existing knowledge. The paper can also strengthen discussion and comparison with related works that have examined the same topic.

---

> ### Author Response · Authors · 2021-11-23
> **response**
>
> Thank you for thoughtful review! We would like to address your concerns with this paper and hope to continue improving on our paper so that the main message is clear and convincing.
>
> ### Paper’s Contributions
>
> It is true that previous studies have shown that low distortion lead to good reconstruction but poor sampling. However, to our knowledge, no paper has explicitly pointed out as we do in section 3.2 that a small amount of information suffices to achieve perceptually good reconstructions (despite poor mean-squared error). This is core to our first message: we do not actually need to model a lot of information to achieve perceptually satisfactory samples.
>
> ### Purpose of Two-Stage VAE
>
> Our main priority is characterizing the trade-off not just between rate-distortion, but also the third dimension: sample quality. The goal of the two-stage VAE is primarily to demonstrate the second message is: when you do try to model a lot of information at once, there is no reason for the prior distribution model to prioritize the perceptually important information. This, we argue, is the reason why high-rate models are likely to generate poor samples (Fig 5). It is not that high-rate is inherently bad, but simply that naively training a high-rate model and hoping for it to be good at sample quality is an unlikely endeavor---akin to finding a needle in a haystack.
>
> Collectively, message #1 and #2 form a testable hypothesis that that high-rate models can be guided to find the “needle in the haystack”. The goal of the two-stage training procedure is thus a demonstration of the hypothesis: showing that we can indeed guide the training of high-rate models to achieve both good ELBOs, and good sample quality.
>
> This insight, we believe, is important for deepening our understanding of sample quality in VAEs and will lead to the development of better generative models in the future.
>
> ### Systematic Experimental Design and Comparison to Prior Work
>
> While we believe that the experiments in our paper are sufficient in demonstrating the core of our message, we appreciate the reviewer’s concern regarding careful methodology description and closer connection to existing work. We aim to address these issues with a more methodological comparison with the existing VDVAE and are confident that the message in the paper will still stand. We hope the reviewer will look forward to a future iteration of our paper!

---

> > ### Comment · Reviewer_sh3z · 2021-11-24
> > **Final thoughts**
> >
> > Thanks for providing the clarifications on the focus of the work on sampling quality. I do see the difference, and appreciate the insights provided along that dimensions. However, I think the ability of the model to reconstruct vs. generating samples, i.e., the sampling quality is closely tied to the rate-distortion balance discussed in Alemi et al -- i.e., the sampling quality examined in this paper is not only tied to the bits encoded in the latent space, but also whether the models learned to simply reconstruct, or is able to generate from an isotropic prior. Like the author suggested, this paper examines the third dimension of the relationship -- the discussion of a third dimension therefore cannot be in isolation from the other dimensions.
> >
> > This seems to be a common concern from myself and Reviewer 7Pio. I echo reviewer 7Pio's suggestion that a future version of the paper can be improved by putting the paper's investigation in the context of what already discussed in Alemi et al, or at least a careful discussion on that matter.
> >
> > I look forward to an improved iteration of this work. For the current form of the manuscript, I will stay with my current rating.

---

### Official Review · Reviewer_Vnek · 2021-11-03

**Correctness:** 3
**Technical Novelty And Significance:** 3
**Empirical Novelty And Significance:** 4
**Recommendation:** 6
**Confidence:** 3

**Main Review:**

This paper is quite good and I'm very excited by the idea.  Reconstruction losses have been a huge part of neural networks going back several decades, and they've always performed somewhat poorly, despite a great deal of research.

The idea in this paper, that we can reframe reconstruction losses as a constraint, is intriguing.  At the same time, I feel that the constraint formulation in equations 3/4 feels a little bit off to me, since it seems like there will be some examples which we can reconstruct perfectly, so it feels odd to force them to have reconstruction error of \gamma.  Maybe if the data is homogeneous enough we can simply find a \gamma which works for all examples, but I think this will be problematic if the data is diverse.  For example if some images are solid color, I see no reason why I need to have a small error \gamma instead of letting these examples have perfect reconstructions.

I also think this paper's presentation is too limiting, in that the ideas here are clearly applicable everywhere that we use reconstruction errors and not just VAEs.  However this makes the presentation more concrete, so I suppose it's okay.

I also wonder how much the visually imperceptible details can be removed simply by lowering the resolution (and then up-sampling).  I suspect this is imperfect and does remove more important information than the constrained formulation, but it feels like a good baseline to study.  I also think that it reflects one of the major ways that people have tried to make reconstruction losses perform better, is to do modeling at a low resolution in the first stage and then a higher resolution at the next stage.

Small Comments:
  -Section 3.1 is very nicely written but it's not clear to me that it's essential to the actual story.

  -It's just a matter of style, but I feel like Figures 4 and Figures 5 are the most striking, so using them to construct the teaser figure at the start of the paper would be compelling.


**Summary Of The Paper:**

This paper argues that much of the likelihood in a VAE is taken up by visually imperceptible information.  Thus a VAE may prefer to model things which are unimportant before learning important visual details.  The authors propose to address this by learning a multi-stage model where the first stage learns coarse visual details and the second stage learns to produce original images.  The technique for learning the first stage involves reframing reconstruction error as a constrained optimization problem where the reconstruction error is not reduced to zero but rather is reduced to some hyperparameter value \gamma.

The improvement in visual samples is impressive, especially on CelebA.

**Summary Of The Review:**

This paper deals with a decades old question in neural networks research, of how we can produce high-quality outputs using reconstruction errors.  Their solution is logical but I have some questions about the details.  Nonetheless I think it represents progress on this problem and should be accepted.  I also found the samples to be impressive.  I believe that this paper will have very high impact.

---

> ### Author Response · Authors · 2021-11-23
> **response**
>
> Thank you for thoughtful review! We would like to address your concerns with this paper and hope to continue improving on our paper so that the main message is clear and convincing.
>
> ### Use of a reconstruction target
>
> Targeting models on the rate-distortion curve as been performed in previous papers, notably by Alemi et al. Regarding your concern about diverse data, please note that the reconstruction target is not performed in a per-sample manner---in other words, we only enforce that the average reconstruction across the entire dataset achieves a particular value.
>
> Where we differ from existing rate-distortion targeting papers is how we applied rate-distortion targeting to investigate a particular phenomenon. Whereas Alemi et al characterizes the trade-off between rate and distortion, our goal is to extend this characterization to a third dimension: sample quality. We wish to note that Eq (3-4) is not the claim to novelty in the paper. It is, after all, simply a tool for targeting a particular rate-distortion value. The novelty in our work is in showing that:
> 1. We do not actually need to model a lot of information to achieve perceptually satisfactory samples (Fig 4)
> 2. When you do try to model a lot of information at once, there is no reason for the prior distribution model to prioritize the perceptually important information (Fig 5).
>
> Collectively, we interpret these findings to mean that it is not that high-rate is inherently bad, but simply that naively training a high-rate model and hoping for it to be good at sample quality is an unlikely endeavor---akin to finding a needle in a haystack.
>
> Subsequently, the goal of the two-stage training procedure is to demonstrate that we can indeed guide the training of high-rate models to achieve both good ELBOs and good sample quality, thereby finding the needle in the haystack. To our knowledge, no existing work specifically highlighted and experimentally demonstrated this line of reasoning.
>
> ### Relation to resolution-reduction
>
> It is certainly true that reducing the resolution (especially if the initial data distribution is very high-resolution) would also be a viable way to remove perceptually less-relevant information. Practically speaking, however, resolution reduction faces two notable challenges. First, we are restricted to a finite number of down-resolution operations and requires modification of the architecture, whereas the selection of a reconstruction target is considerably simpler. Second, the model may still naively choose to target a high-rate in the reduced-resolution dataset by minimizing the reconstruction error toward zero on the reduced resolution dataset. Directly targeting a reconstruction value avoids this potential issue. Because of this, we found it most natural to study how VAEs behave in terms of sample quality along the rate-distortion curve in particular.

---

### Decision · Program_Chairs · 2022-01-20

**Decision:**

Reject

**Comment:**

The authors discuss the disconnect between log-likelihood and sample quality of VAEs and relate it to an undesirable focus of the model on high-frequency signals. They propose to alleviate it through a two-stage training scheme for VAEs.
As it is, the paper does not explain well its contributions, especially compared to the rate-distortion balance discussion in "Fixing a Broken ELBo" by Alemi et al. (2018) (see [reviews sh3z](https://openreview.net/forum?id=-0LuSWi6j4&noteId=D52ninjThn1), [7Pio](https://openreview.net/forum?id=-0LuSWi6j4&noteId=9qMQNUGk6bx), and [LBJj](https://openreview.net/forum?id=-0LuSWi6j4&noteId=gyG86hghxsU)), and lacks the experiments to back up its claim (see [LBJj](https://openreview.net/forum?id=-0LuSWi6j4&noteId=gyG86hghxsU), and [KKon](https://openreview.net/forum?id=-0LuSWi6j4&noteId=zeFApaHliSv)). While the authors have made a more precise statement about their contributions in their rebuttal, the writing remains unclear.
I recommend this submission for rejection.